# ON THE ROLE OF DRUG REPRESENTATIONS IN SINGLE-CELL PERTURBATION MODELING

**Marco Pegoraro**[*,3]**, Sanketh Vedula**[*,1,4,5†]**, Lion Halika**[1]**, Michael M. Danziger**[2]**,
Alex Bronstein**[1,3]**, Michal Rosen-Zvi**[2]
[1]Technion, Israel  [2]IBM Research, Israel  [3]IST Austria
[4]Princeton University  [5]Broad Institute of MIT and Harvard
[*]Equal contribution

## ABSTRACT

Predicting cellular responses to small-molecule perturbations is a central challenge in computational biology and drug discovery. Although recent single-cell foundation models learn rich representations of cellular state from large transcriptomic datasets, their performance on drug perturbation prediction remains limited. This raises an important question as to whether current shortcomings arise from how drugs are represented, how cells are represented, or how the two are coupled. We leverage the Tahoe-100M dataset, which contains single-cell perturbation screens across approximately 50 cell lines and 350 drugs, to study this question at scale. We construct biological-response–derived drug embeddings (BiRD embeddings) from transcriptional response similarities and show that they capture biologically relevant variation not captured by chemical-structure descriptors or small-molecule foundation models such as ADMET-AI. We then introduce an optimal transport–based objective to align single-cell foundation model representations with the BiRD space. After fine-tuning, models achieve large gains on a perturbation-retrieval task, reaching 0.8–0.9 AUC compared to 0.5–0.6 AUC with ADMET-based embeddings, and generalize to unseen cell lines. Together, these results highlight the importance of grounding drug representations in biological response data for accurate and transferable perturbation modeling.

## 1 INTRODUCTION

Predicting how cells respond to external perturbations is a central goal of computational biology, with direct implications for drug discovery, functional genomics, and precision medicine. In particular, accurately predicting transcriptional responses to small-molecule perturbations could enable *in silico* screening of candidate therapeutics and systematic characterization of cellular pathways. Despite rapid progress in large-scale single-cell transcriptomics, reliably predicting perturbation responses remains a challenging and largely unsolved problem.

Recent years have seen the emergence of single-cell foundation models trained on millions of transcriptomes, including models such as scGPT (Cui et al., 2024), Geneformer (Theodoris et al., 2023), BMFM (Dandala et al., 2025), and related architectures that learn general representations of cellular state. While these models capture broad biological structure, their performance on perturbation prediction tasks has been mixed. Several studies report that simple linear or task-specific models can match or outperform large pretrained models on downstream perturbation benchmarks (Baek et al., 2025; Wu et al., 2025), with transferring perturbation predictions to unseen cell lines as a particularly challenging task (Li et al., 2025).

A growing body of work has begun to address post-perturbation prediction directly. Methods such as CellOT (Bunne et al., 2023) and BioLORD (Piran et al., 2024) model cellular transitions between pre- and post-perturbation states, typically focusing on learning mappings in gene-expression space. However, these approaches largely treat drug identity as fixed input features. In practice, most methods rely on chemical fingerprints or emerging small-molecule foundation models (e.g.,

---

[†]Work done while affiliated with Technion

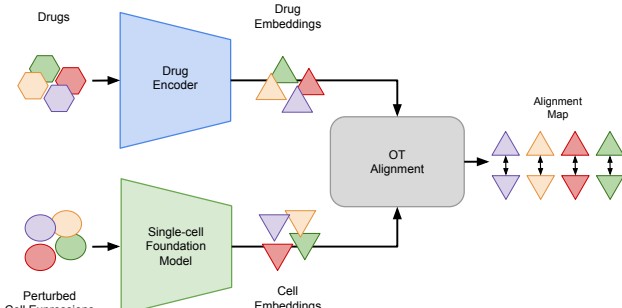

Figure 1: **The proposed fine-tuning strategy.** Drug and perturbed cell expression inputs are encoded into separate embedding spaces. An entropically regularized OT objective learns a soft alignment map between the two modalities, enforcing consistency between drug representations and cellular perturbation states.

ADMET-AI (Swanson et al., 2024)) to represent compounds. Yet, it remains unclear whether chemically derived embeddings capture the aspects of drug similarity that are most relevant to cellular response. This motivates a central question: *are current limitations in perturbation prediction driven by how we represent cells, how we represent drugs, or how the two representations interact?*

The recent availability of large-scale single-cell perturbation datasets provides an opportunity to revisit this question. In particular, Tahoe-100M contains perturbational transcriptomic profiles spanning 50 cell lines and 380 drugs. This substantially expands the scale of prior resources such as sci-Plex (188 drugs across three cell lines (Srivatsan et al., 2020)) and L1000, which relied on marker-gene microarrays rather than full transcriptomic measurements (Subramanian et al., 2017).

**Contributions.** Leveraging these developments, we introduce *biological-response–derived drug embeddings (BiRD)*. Unlike chemical or structure-based representations, BiRD embeddings are constructed directly from transcriptional responses measured in a set of reference cell lines. This yields drug representations that explicitly encode how compounds perturb gene-expression programs.

Building on this representation, we further propose a finetuning strategy that makes single-cell foundation models *drug-aware*. Specifically, we introduce an optimal transport–based objective that aligns the embedding space of single-cell foundation models with the BiRD drug embedding space. This promotes the metric in the expression space to be aligned with the drug embedding space, enabling the model to generalize to unseen compounds.

Our central hypothesis is that grounding drug representations in biological response data will produce higher-quality embeddings for downstream perturbation prediction and improve the effectiveness and transferability of single-cell foundation models. To evaluate this hypothesis, we design two complementary tasks. First, we introduce a *k-NN post-perturbation prediction* task. This task isolates the quality of drug embeddings by predicting the transcriptional response of a drug using only its nearest neighbors in the embedding space, without training any predictive model. Second, we introduce a *perturbation retrieval* task in which a perturbed cell embedding from a single-cell foundation model must be matched to the correct drug, including drugs not observed during training.

Our results suggest that BiRD embeddings substantially improve post-perturbation expression prediction, consistently yielding lower MAE and outperforming chemical fingerprints and small-molecule foundation models in pairwise comparisons (Figure 3). When used to finetune single-cell FMs, BiRD enables strong perturbation retrieval performance (AUC: 0.8-0.9 vs. 0.5-0.6 for the baselines) and demonstrates better generalization across unseen cell lines. These results suggest that biologically grounded drug representations are likely an important ingredient for solving perturbation prediction, while also indicating that existing drug representations, although useful, may not yet fully capture the aspects of drug similarity most relevant to cellular response.

**Related work.** ChemCPA (Hetzel et al., 2022) is a closely related attempt to make perturbation models generalize to unseen compounds by learning drug embeddings from chemical structure *end-to-end*: a molecule encoder produces a drug representation that is mapped into an additive "perturbation" latent, and the resulting manifold is shaped implicitly by optimizing post-perturbation reconstruction. In contrast, our approach fixes the drug geometry *a priori* by constructing BiRD embeddings directly from measured transcriptional responses in reference cell lines, yielding a biologically grounded notion of drug similarity independent of any predictor. This decoupling lets us

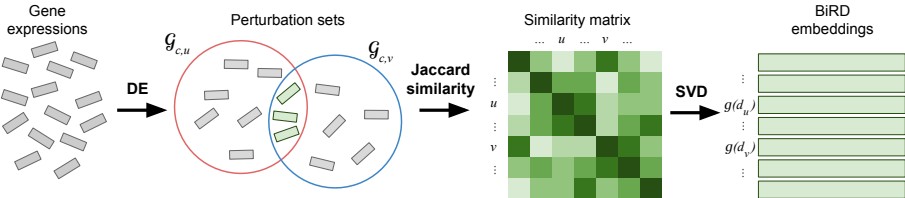

Figure 2: **Biological-response–derived Drug Embeddings (BiRD).** Starting from gene expression profiles, we perform differential expression (DE) analysis to identify drug-specific perturbation gene sets. We then compute the pairwise Jaccard similarity between these perturbation sets to construct a drug–drug similarity matrix. Finally, we obtain low-dimensional drug embeddings by applying spectral decomposition (SVD) to this similarity matrix.

evaluate drug representations in isolation and then align single-cell foundation model embeddings to a response-derived drug metric, rather than relying on a learned chemical-to-phenotype geometry.

**Notation.** We consider a dataset of single-cell cellular states $\mathcal{X} = \{\mathbf{x}_i\}_{i=1}^N$, where each cell $\mathbf{x}_i$ has been perturbed by a drug from a set $\mathcal{D} = \{d_1, \ldots, d_K\}$. We define a mapping function $k(i) \in \{1, \ldots, K\}$ which returns the index of the drug applied to cell $i$. We define $g : \mathcal{D} \to \mathbb{R}^L$ as the drug embedding function mapping discrete drug identities to continuous vectors, and $f_\theta : \mathcal{X} \to \mathbb{R}^L$ as the cellular embedding function parameterized by $\theta$. Both functions project their inputs into a common latent space $\mathbb{R}^L$, where we perform alignment.

## 2 BIOLOGICAL-RESPONSE–DERIVED DRUG EMBEDDINGS

We introduce **BiRD**, a drug embedding designed to capture the functional relationships between drugs. We assume access to transcriptomic perturbation data from a set of reference cell-lines ($\mathcal{C}_{\mathrm{ref}}$), specifically A549 (Lung), AN3 CA (Uterus), SK-MEL-2 (Skin), SNU-1 (Esophagus/Stomach), and A-172 (CNS). Crucially, these reference contexts are strictly excluded from the training, validation, and test sets used for the downstream single-cell foundation model fine-tuning to ensure no data leakage. The overall BiRD pipeline is illustrated in Fig. 2.

**Differential-response signatures.** We first characterize the functional footprint of each drug $d_k \in \mathcal{D}$ by extracting differential expression (DE) signatures. To minimize technical variation, DE is computed relative to unperturbed control cells from the same experimental plate. Within each reference context $c \in \mathcal{C}_{\mathrm{ref}}$, we capture the most significant transcriptomic perturbations by filtering for statistical significance ($q_{c,k} \leq \alpha$) and retaining only the set $\mathcal{G}_{c,k}$ of the top $M$ genes with the largest absolute log-fold change $|\mathbf{\Delta}_{c,k}|$.

**Drug similarity.** We quantify pairwise drug proximity using the Jaccard index of these perturbation sets. We construct a context-specific similarity matrix $\mathbf{S}^{(c)} \in \mathbb{R}^{K \times K}$ where the entry for drugs $d_u$ and $d_v$ is:

$$\mathbf{S}_{uv}^{(c)} = \frac{|\mathcal{G}_{c,u} \cap \mathcal{G}_{c,v}|}{|\mathcal{G}_{c,u} \cup \mathcal{G}_{c,v}|}, \quad \text{with } \mathbf{S}_{uu}^{(c)} = 1. \tag{1}$$

The similarity metric is invariant to absolute expression levels, and the resulting graph topology is driven by shared gene identities rather than batch-dependent variations in response magnitude.

**Spectral embedding.** To resolve the global structure of drug relationships into embeddings, we compute the eigendecomposition of the similarity matrix $\mathbf{S}^{(c)} = \mathbf{Q}\mathbf{\Lambda}\mathbf{Q}^\top$. To construct the $L$-dimensional embedding $\mathbf{E}^{(c)} \in \mathbb{R}^{K \times L}$, we retain the $L$ leading eigenvalues and their corresponding eigenvectors. Formally, we define $\mathbf{E}^{(c)} = \mathbf{Q}_L \mathbf{\Lambda}_L^{1/2}$, where $\mathbf{\Lambda}_L = \mathrm{diag}(\lambda_1, \ldots, \lambda_L)$ contains the largest eigenvalues and $\mathbf{Q}_L$ contains the associated eigenvectors. When multiple reference contexts are available, we concatenate these local embeddings to form a unified representation $\mathbf{E} = [\mathbf{E}^{(c_1)} \| \ldots \| \mathbf{E}^{(c_{|\mathcal{C}_{\mathrm{ref}}|})}]$. Finally, we define the drug embedding $g(d_k)$ as the $k$-th row of $\mathbf{E}$.

## 3    DRUG-AWARE FINETUNING OF SINGLE-CELL FMS

While single-cell Foundation Models (FMs) learn general cellular representations, they typically lack explicit knowledge of drug perturbations. As a result, fine-tuning is often required to adapt these models to perturbation-specific tasks. Our goal is to equip single-cell FMs with drug awareness by aligning perturbed cellular representations with the intrinsic geometry of the drug embeddings. Given a training dataset consisting of post-perturbation cell states and their corresponding drug embeddings, we fine-tune the FM to reduce the discrepancy between the transcriptomic representation space and the drug embedding space, encouraging consistency between cellular responses and drug-level structure. Fig. 1 provides an overview of the proposed fine-tuning strategy.

Given the fixed drug embeddings $g(d_k)$, we learn the cellular embedding function $f_\theta$ by aligning the resulting cell embeddings with the drug embeddings.

We define the empirical marginals for the cells ($a$) and the drugs ($b$) as:

$$a_i = \frac{1}{N} \quad \text{and} \quad b_k = \frac{N_k}{\sum_{\ell=1}^{K} N_\ell},$$

where $N_k$ denotes the count of cells perturbed by drug $d_k$ in the batch.

We define a quadratic ground cost $\mathbf{C} \in \mathbb{R}^{N \times K}$ representing the distance between a cell and a drug in the latent space:

$$\mathbf{C}_{ik} = \|f_\theta(\mathbf{x}_i) - g(d_k)\|_2^2.$$

We compute the *entropy-regularised optimal (OT) transport* (Cuturi, 2013) plan $\hat{\mathbf{\Pi}}$ by solving:

$$\hat{\mathbf{\Pi}} = \arg \min_{\mathbf{\Pi} \in U(a,b)} \langle \mathbf{\Pi}, \mathbf{C} \rangle - \epsilon H(\mathbf{\Pi}), \tag{2}$$

where the feasible set of transport plans is defined by the marginal constraints:

$$U(a,b) = \left\{ \mathbf{\Pi} \in \mathbb{R}_+^{N \times K} : \mathbf{\Pi} \mathbf{1}_K = a, \ \mathbf{\Pi}^\top \mathbf{1}_N = b \right\},$$

and $H(\mathbf{\Pi}) = -\langle \mathbf{\Pi}, \log \mathbf{\Pi} \rangle$ is the entropic regularization with coefficient $\epsilon > 0$.

Finally, we optimize the alignment objective:

$$\mathcal{L}_{\text{OT}} = -\langle \mathbf{\Pi}^\star, \log \hat{\mathbf{\Pi}} \rangle = -\frac{1}{N} \sum_{i=1}^{N} \log \hat{\mathbf{\Pi}}_{i,k(i)}, \tag{3}$$

where $\hat{\mathbf{\Pi}}$ denotes the optimal transport plan predicted by eq. (2), and $\mathbf{\Pi}^\star$ is the ground-truth assignment matrix induced by the mapping function $k(i)$, which maps each cell $i$ to the index of the drug $d_{k(i)}$ that perturbed it. Importantly, this assignment is not a permutation matrix: a single drug $d_k$ may correspond to multiple cellular states $\mathbf{x}_i$, resulting in a many-to-one mapping from cells to drugs. The entropy-regularised OT problem is solved using the Sinkhorn algorithm (Cuturi, 2013), implemented in the log-domain for numerical stability. During training, gradients are propagated through a fixed number of Sinkhorn iterations, enabling end-to-end optimisation of the cellular encoder (Eisenberger et al., 2022).

## 4    EVALUATION TASKS

We evaluate the utility of drug embeddings through two complementary tasks that probe different failure modes. The first assesses the *intrinsic quality of the drug embedding space* by asking whether distances in $g(\cdot)$ alone predict transcriptional outcomes, independent of any cell encoder. The second assesses the *alignment between a single-cell foundation model and the drug embedding space* by testing whether perturbed cells can retrieve their causative drug in a zero-shot setting. Accordingly, we report k-NN post-perturbation prediction accuracy (global MAE, win-rates, and gene-level enrichment of gains), followed by zero-shot retrieval and embedding separability metrics.

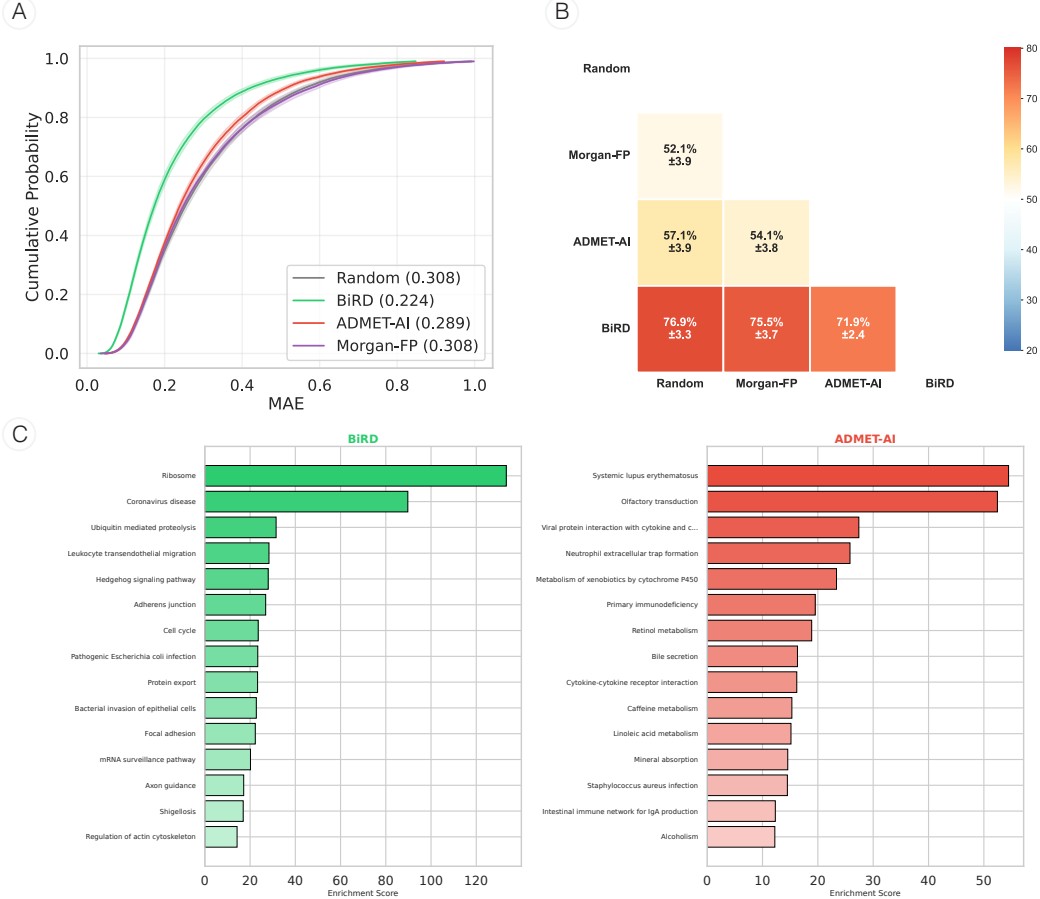

Figure 3: **BiRD embeddings improve post-perturbation expression prediction.** We evaluate drug embeddings by predicting post-perturbation transcriptional responses: for each drug, the response is estimated as the average response of its nearest neighbors in the embedding space (BiRD, ADMET-AI, Morgan-FP, and a random baseline). (A) Empirical CDF of mean absolute error (MAE) across drugs, with shaded confidence intervals computed across 45 cell lines. BiRD consistently shifts the error distribution toward lower MAE, indicating more accurate prediction across the full drug set. (B) Pairwise win-rate matrix showing the percentage of drug–cell-line settings in which the method in each row achieves lower MAE than the method in each column. BiRD outperforms all alternatives by a large margin, while ADMET-AI modestly outperforms Morgan-FP, which only slightly improves over the random baseline. (C) Gene-level analysis of performance gains. For each method, we identify genes with the largest MAE reductions and perform GO enrichment on the resulting gene sets. BiRD-associated genes show enrichment for pathway-level and systems-biology processes (e.g., ribosome, cell cycle, adhesion and signaling pathways), whereas ADMET-AI gains are enriched for metabolism, absorption/secretion, immune, and signaling pathways.

**k-NN post-perturbation prediction.** This task isolates the quality of the drug embeddings $g(\cdot)$ by predicting the transcriptional response of a held-out drug solely based on its position in the drug manifold. We assume access to a set of training drugs $\mathcal{D}_{\text{train}}$ for which we have observed post-perturbation cellular responses, and a set of held-out test drugs $\mathcal{D}_{\text{test}}$. For a query drug $d_{\text{test}} \in \mathcal{D}_{\text{test}}$, let $\mathcal{N}_k(d_{\text{test}}) \subset \mathcal{D}_{\text{train}}$ denote its $k$ nearest neighbors in the drug embedding space under Euclidean distance $\|g(d_{\text{test}}) - g(d)\|_2$. We predict the transcriptional profile of $d_{\text{test}}$ by aggregating the mean expression profiles of cells perturbed by its neighbors:

$$\hat{\boldsymbol{\mu}}_{d_{\text{test}}} = \frac{1}{|\mathcal{N}_k(d_{\text{test}})|} \sum_{d_j \in \mathcal{N}_k(d_{\text{test}})} \left( \frac{1}{|\mathcal{X}_{d_j}|} \sum_{\mathbf{x} \in \mathcal{X}_{d_j}} \mathbf{x} \right), \tag{4}$$

where $\mathcal{X}_{d_j}$ is the set of observed cells perturbed by drug $d_j$. Performance is measured using mean absolute error (MAE) between $\hat{\boldsymbol{\mu}}_{d_{\text{test}}}$ and the ground-truth mean expression of cells perturbed by $d_{\text{test}}$, computed per drug–cell-line setting and aggregated across cell lines. We additionally report (i) pairwise win-rates, defined as the fraction of drug–cell-line settings where one embedding yields lower MAE than another, and (ii) a gene-level analysis based on per-gene MAE deltas and downstream pathway enrichment to characterize which transcriptional programs are captured by each drug representation. This task therefore measures the degree to which proximity in drug embedding space corresponds to similarity in transcriptional response.

**Contextual zero-shot perturbation retrieval.** This task evaluates the alignment between the cellular embedding space induced by the single-cell foundation model $f_\theta$ and the drug embedding space induced by $g$. Given a query cell $\mathbf{x}_{\text{query}}$ perturbed by a drug $d_{\text{target}} \in \mathcal{D}_{\text{test}}$ that is unseen in the current cell line, the goal is to identify $d_{\text{target}}$ from the candidate set of all drugs $\mathcal{D}$. We score each candidate drug by its distance to the cellular embedding and retrieve the nearest drug embedding:

$$\hat{d} = \arg\min_{d \in \mathcal{D}} \|f_\theta(\mathbf{x}_{\text{query}}) - g(d)\|_2. \tag{5}$$

To quantify performance, we treat retrieval as a one-vs-all perturbation identification problem and report micro-averaged AUROC over drugs using scores $s(d \mid \mathbf{x}_{\text{query}}) = -\|f_\theta(\mathbf{x}_{\text{query}}) - g(d)\|_2$, evaluated on held-out drugs (contextual zero-shot). Crucially, this setup assumes that while the drug is unseen in the *target* cell line, its embedding $g(d)$ has been pre-computed using transcriptional signatures from the reference cell lines. This setup simultaneously assesses the quality of these reference-derived embeddings and the extent to which the single-cell FM has learned to transfer transcriptomic shifts to the correct perturbation representation. In the Appendix A, we report additional implementation details.

## 5 RESULTS

We evaluate drug representations using two complementary tests introduced above. First, we measure whether the geometry of the drug embedding space alone supports prediction of post-perturbation transcriptional responses via neighborhood aggregation (k-NN prediction). Second, we assess whether the single-cell foundation model can be aligned to a given drug space so that perturbed cells retrieve their causative drug in a zero-shot setting (perturbation retrieval). Unless otherwise noted, results are aggregated across the 45 evaluation cell lines. To ensure rigorous evaluation, the five reference aforementioned cell lines used to construct the BiRD embeddings were strictly excluded from both the k-NN prediction and zero-shot retrieval tasks. Consequently, performance metrics reflect generalization to cellular contexts that were completely unseen during the construction of the drug embeddings.

### 5.1 K-NN POST-PERTURBATION PREDICTION

**BiRD improves response prediction across the full drug set.** We first test whether distances in drug embedding space are predictive of post-perturbation expression. For each test drug $d \in \mathcal{D}_{\text{test}}$ (and cell line $c$), we predict its mean post-perturbation profile by averaging the observed mean profiles of its $k$ nearest training drugs under the embedding (Task 1). We compare BiRD against ADMET-AI embeddings, Morgan fingerprints (Morgan-FP), and a random embedding baseline, and

evaluate performance using mean absolute error (MAE) between predicted and ground-truth mean expression profiles.

Across all global summaries, BiRD achieves the lowest error. The empirical CDF of MAE (Fig. 3A) shows a consistent shift toward lower errors, indicating improved prediction across the majority of drugs (and drug–cell-line settings). Pairwise win-rate analysis (Fig. 3B) corroborates this trend: BiRD yields lower MAE than each competing representation in the majority of $(d, c)$ settings. In contrast, ADMET-AI provides only modest gains over Morgan-FP, and both are only marginally better than the random baseline.

**Gene-level error decomposition and functional enrichment.** To contextualize the global differences in MAE, we next analyze prediction error at the gene level. For each gene, we compute its MAE under each drug representation (aggregated across drug–cell-line settings) and quantify method-specific differences via per-gene MAE deltas (e.g., $\Delta \text{MAE}_{\text{BiRD-ADMET}}$). We then rank protein-coding genes by these deltas and perform Gene Ontology enrichment on the top 500 genes exhibiting the largest error reductions for each method (Fig. 3C). This gene-level decomposition provides a complementary, functionally interpretable view of what biological processes are emphasized by different drug embedding spaces.

Genes with the largest BiRD-associated error reductions are enriched for pathway-level processes including ribosome, cell cycle, adhesion, and signaling, consistent with improved modeling of coordinated transcriptional programs induced by perturbations. In contrast, genes preferentially improved by ADMET-AI are enriched for metabolism, absorption/secretion, immune, and related signaling pathways. The enrichment of metabolism-associated terms for ADMET-AI is consistent with its training objective, which emphasizes pharmacokinetic and drug-property signals that are closely related to metabolic processing and transport. Together, these trends suggest that biologically grounded embeddings better capture broad transcriptional programs, whereas property-based embeddings primarily reflect pharmacokinetic and metabolism-linked effects.

## 5.2 ZERO-SHOT PERTURBATION RETRIEVAL

**BiRD yields substantially stronger perturbation identification.** We next evaluate alignment quality by asking whether perturbed cells retrieve their causative drug embedding (Task 2). Given a query cell $\mathbf{x}_{\text{query}}$ perturbed by an unseen drug, we score each candidate drug by its proximity to the cell embedding and compute micro-averaged one-vs-all AUROC over drugs (Appendix A). As shown in Fig. 4, BiRD achieves the strongest performance with an average AUROC of $0.92 \pm 0.04$, substantially outperforming Morgan-FP ($0.60 \pm 0.03$) and ADMET-AI ($0.57 \pm 0.03$). These results indicate that BiRD provides a more informative target space for mapping transcriptomic shifts to perturbation identity in a zero-shot regime.

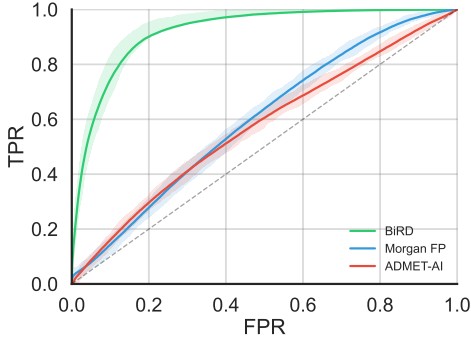

Figure 4: **ROC curves for the perturbation prediction task.** BMFM is fine-tuned with three different drug embedding types, and ROC curves are averaged across 48 cell lines.

**BiRD alignment is more transferable across cell lines.** We further examine generalization across cellular contexts. In Appendix C.1, we report cross-cell-line experiments showing that models fine-tuned using BiRD maintain strong performance when evaluated on held-out cell lines, supporting the interpretation that BiRD captures perturbation structure that transfers across cell types.

**Fine-tuning improves drug separability in the cellular embedding space.** Finally, we assess whether the proposed OT-based fine-tuning improves the organization of the cellular embedding space with respect to drug labels. Using scIB separability metrics computed on test-set embeddings and averaged across models trained on different cell lines, we observe consistent improvements over the frozen BMFM baseline (Table 1). The aggregated score increases from $0.79 \pm 0.03$ (frozen

BMFM) to 0.84–0.87 after fine-tuning, with BiRD achieving the highest overall performance ($0.87\pm0.02$). This indicates that alignment not only improves retrieval accuracy but also yields a cellular representation where drug perturbations are more cleanly separable.

## 6 CONCLUSION

These results highlight the substantial performance gap between representations derived solely from chemical structure and those grounded in biological observation. Chemical priors, distinct from the biological priors used in BiRD, often lack the resolution to distinguish functional nuances in transcriptional response. We recognize that capturing this resolution comes with a prerequisite: unlike purely in silico methods like ChemCPA or ADMET-AI, our approach relies on screening drugs in reference cell lines. Yet, our data demonstrates that this step is crucial for constructing representations that truly reflect functional perturbations. By prioritizing biological grounding over theoretical scalability, BiRD achieves the high-fidelity generalization needed to model unseen cellular contexts.

### MEANINGFULNESS STATEMENT

Understanding life requires representations that reflect how biological systems actually respond to perturbation. Our work demonstrates that models grounded in observed transcriptional responses capture functional relationships that purely structure-based priors miss. By anchoring representations in biological measurement rather than chemical abstraction alone, BiRD learns embeddings that encode cellular function, not just molecular form. In doing so, our approach advances the goal of learning representations that are not only predictive, but biologically faithful.

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

---

**Algorithm 1 BiRD:** Biological-response–derived Drug Embeddings

---

**Input:** Expression matrix $\mathbf{X}$, Metadata $(d(i), c(i))$, Ref. cell lines $\mathcal{C}_{\text{ref}}$
**Input:** Signature size $K$, Embedding dimension $k$
**Output:** Drug embeddings matrix $\mathbf{E}$

**Stage 1: Construct differential-response signatures**
1: **for** each cell line $c \in \mathcal{C}_{\text{ref}}$ and drug $d \in \mathcal{D}$ **do**
2:      Compute DE stats (LogFC $\boldsymbol{\Delta}_{c,d}$, $q$-values)           $\triangleright$ Treated vs. Control
3:      $\mathcal{H}_{c,d} \leftarrow \{j : q_{c,d}(j) \leq \alpha\}$           $\triangleright$ Filter significant genes
4:      $\mathcal{G}_{c,d} \leftarrow \text{TopK}\left(|\boldsymbol{\Delta}_{c,d}(j)|, \ j \in \mathcal{H}_{c,d}\right)$           $\triangleright$ Select top $K$ responsive genes
5: **end for**

**Stage 2: Transcriptomic-response drug similarity**
6: **for** each cell line $c \in \mathcal{C}_{\text{ref}}$ **do**
7:      Compute Jaccard similarity matrix $\mathbf{S}^{(c)}$:
8:      $\mathbf{S}_{ij}^{(c)} \leftarrow \dfrac{|\mathcal{G}_{c,d_i} \cap \mathcal{G}_{c,d_j}|}{|\mathcal{G}_{c,d_i} \cup \mathcal{G}_{c,d_j}|}, \quad \forall d_i, d_j \in \mathcal{D}$
9:      $\mathbf{S}_{ii}^{(c)} \leftarrow 0$           $\triangleright$ Zero diagonal for spectral analysis
10: **end for**

**Stage 3: Spectral embedding of graphs**
11: **for** each cell line $c \in \mathcal{C}_{\text{ref}}$ **do**
12:      Decompose $\mathbf{S}^{(c)} \approx \mathbf{Q}^{(c)} \boldsymbol{\Lambda}^{(c)} (\mathbf{Q}^{(c)})^{\top}$           $\triangleright$ Top-$k$ eigenpairs
13:      $\mathbf{E}^{(c)} \leftarrow \mathbf{Q}_{:,1:k}^{(c)} (\boldsymbol{\Lambda}_{1:k}^{(c)})^{1/2}$           $\triangleright$ Generate local embeddings
14: **end for**

**Stage 4: Fusing embeddings across reference cell-lines**
15: $\mathbf{E} \leftarrow \left[\, \mathbf{E}^{(c_1)} \parallel \mathbf{E}^{(c_2)} \parallel \cdots \parallel \mathbf{E}^{(c_{|\mathcal{C}_{\text{ref}}|})} \,\right]$           $\triangleright$ Concatenate across cell lines
16: **return** $\mathbf{E}$

---

# A    IMPLEMENTATION

We fine-tune the Biomedical Foundation Model (BMFM) (Dandala et al., 2025) to incorporate drug-level information using the optimal transport (OT) alignment objective defined in eq. (3). We initialise the model with the publicly released pretrained weights from IBM's `biomed-multi-omics` repository on HuggingFace. We freeze the BMFM encoder and introduce a learnable linear projection that maps the BMFM cellular embeddings to a perturbed cellular representation with the same dimensionality as the drug embeddings. We consider three types of drug embeddings: Morgan-FP, ADMET-AI, and BiRD. For each embedding type, we fine-tune a separate model per cell line in the Tahoe-100M dataset (concentration 5), resulting in 48 cell line–specific models. The full list of cell lines and their corresponding tissues is provided in table 2 of the Appendix.

## A.1   DATA PREPARATION

We perform pseudo-bulk aggregation of single-cell expression profiles at the drug–cell line level. For each drug treatment within a given cell line, we first filter out low-quality cells with total counts below a predefined threshold. We then generate pseudo-bulk samples by randomly sampling 500 cells with replacement from the corresponding drug–cell line subset and computing the mean gene expression across the sampled cells. This procedure is repeated 50 times, resulting in multiple pseudo-bulk replicates per drug. The resulting pseudo-bulk profiles are split into training, validation, and test sets based on drug identity using an 80% / 10% / 10% proportion. Consequently, drugs in the test set are never observed during training, enabling evaluation of the model's ability to generalize to unseen drug perturbations.

Table 1: **Aligning single-cell representations with drug embeddings yields a drug-aware embedding space.** The table reports clustering-based label-separation metrics computed on test embeddings, averaged across cell lines (mean ± s.d.). Metrics: **Isolated**—label isolation; **NMI**—cluster agreement; **ARI**—clustering accuracy; **Silhouette**—cluster separation; **Agg.**—overall score.

| Embedding | Isolated | NMI | ARI | Silhouette | Agg. |
|---|---|---|---|---|---|
| *Pretrained* | | | | | |
| BMFM | $0.66 \pm 0.02$ | $0.90 \pm 0.03$ | $0.75 \pm 0.07$ | $0.66 \pm 0.02$ | $0.79 \pm 0.03$ |
| *BMFM fine-tuned with drug embeddings* | | | | | |
| Morgan-FP | $\mathbf{0.72 \pm 0.02}$ | $\mathbf{0.97 \pm 0.02}$ | $0.90 \pm 0.04$ | $\mathbf{0.72 \pm 0.02}$ | $0.86 \pm 0.02$ |
| ADMET-AI | $0.69 \pm 0.01$ | $0.96 \pm 0.02$ | $0.88 \pm 0.05$ | $0.69 \pm 0.01$ | $0.84 \pm 0.02$ |
| BiRD | $\mathbf{0.72 \pm 0.02}$ | $\mathbf{0.97 \pm 0.02}$ | $\mathbf{0.92 \pm 0.04}$ | $\mathbf{0.72 \pm 0.02}$ | $\mathbf{0.87 \pm 0.02}$ |

## B    METRICS

The reported scIB metrics (Luecken et al., 2022) quantify biological separability of the learned embeddings:

- **Isolated Labels** measures how well rare or underrepresented cell populations form distinct clusters, assessing robustness to label imbalance.
- **KMeans NMI** (Normalized Mutual Information) and **KMeans ARI** (Adjusted Rand Index) evaluate agreement between unsupervised KMeans clustering assignments and ground-truth biological labels, measuring how well the embedding space recovers known group structure.
- **Silhouette** (label) quantifies how well samples are separated with respect to their true labels by comparing intra-class compactness to inter-class separation.

Together, these metrics assess whether the embedding preserves biologically meaningful structure while enabling clear separation of perturbation-defined cell states.

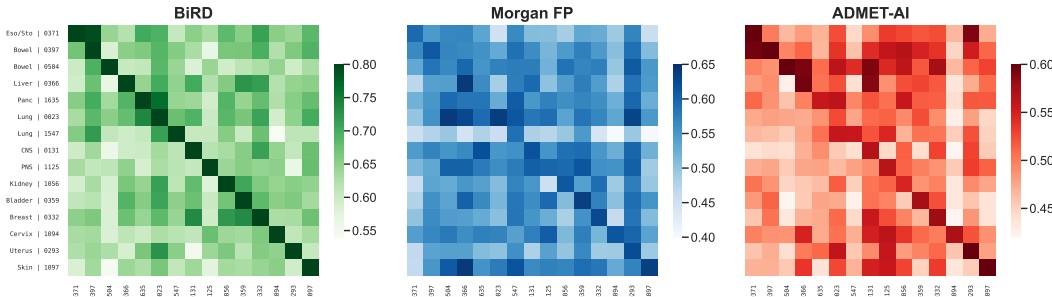

Figure 5: **Generalization performance across cell lines.** AUROC values are reported for fine-tuned models evaluated across different cell lines. Rows indicate the cell line used for fine-tuning, while columns represent the cell lines on which the models were tested.

## C    ADDITIONAL EXPERIMENTS

In this section we report additional experiments.

### C.1    PERTURBATION PREDICTION ACROSS CELL LINES.

BiRD-based fine-tuning yields substantially stronger cross-cell-line generalization compared to alternative drug embeddings. In particular, models trained with BiRD maintain high perturbation prediction performance even when evaluated on unseen cellular contexts.

| Tissue | Cell line IDs (Cellosaurus) |
|---|---|
| Lung | CVCL_1550, CVCL_1731, CVCL_1693, CVCL_1571, CVCL_1531, CVCL_1055, CVCL_1495, CVCL_0023, CVCL_1577, CVCL_1285, CVCL_0459, CVCL_1517, CVCL_1478, CVCL_1716, CVCL_1547 |
| Skin | CVCL_0069, CVCL_1666, CVCL_1097, CVCL_1381 |
| Breast | CVCL_0332, CVCL_0179 |
| Bowel | CVCL_0397, CVCL_0546, CVCL_0504, CVCL_0399, CVCL_0218, CVCL_0292, CVCL_1717, CVCL_1724, CVCL_0320 |
| Esophagus/Stomach | CVCL_0099, CVCL_0371 |
| Pancreas | CVCL_0428, CVCL_1635, CVCL_0152, CVCL_1119, CVCL_0480, CVCL_0334, CVCL_C466 |
| Uterus | CVCL_0028, CVCL_0293 |
| Bladder / Urinary tract | CVCL_0359 |
| CNS / Brain | CVCL_0131, CVCL_1239, CVCL_1715 |
| Liver | CVCL_0366, CVCL_1098 |
| Cervix | CVCL_1094 |
| Peripheral nervous system | CVCL_1125 |
| Kidney | CVCL_1056 |

Table 2: **Cell lines used from the Tahoe-100M perturbation dataset grouped by tissue of origin.** Cell line identifiers follow the Cellosaurus (CVCL) nomenclature.

Figure 5 illustrates cross-cell-line generalization across a subset of cell lines from different tissues. Each row corresponds to the cell line used for fine-tuning, while each column denotes the evaluation cell line. AUROC values are computed on the test set. While all models achieve their best performance along the diagonal (i.e., when trained and tested on the same cell line), clear differences emerge in the off-diagonal entries. BiRD models exhibit consistently high AUROC values across cell lines, with substantially smaller performance degradation compared to Morgan-FP and ADMET-AI. In contrast, Morgan-FP and ADMET-AI models show pronounced drops in AUROC under cross-cell-line transfer.

These results indicate that BiRD embeddings capture perturbation effects that are less cell-line specific and more transferable, leading to improved robustness across diverse cellular contexts.

