# OpenReview forum: "On the role of drug representations for single-cell perturbation modeling"
_ICLR.cc/2026/Workshop/LMRL — ICLR 2026 Workshop LMRL Poster_

### Official Review · Reviewer_fPUn · 2026-02-25
**Well-scoped question, strong execution, but the informational advantage over baselines warrants explicit framing.**

**Rating:** 7
**Confidence:** 4

**Review:**

## 1. Summary

This paper investigates the role of drug representations in single-cell perturbation modeling. The authors introduce BiRD (Biological-response–derived Drug Embeddings), constructed by computing Jaccard similarity over differential expression gene sets across reference cell lines, followed by a spectral embedding of the drug matrix. They then propose an optimal transport–based fine-tuning objective to align single-cell foundation model (BMFM) embeddings with the BiRD drug space. Evaluation is conducted on the Tahoe-100M dataset (~50 cell lines, ~350 drugs) across two tasks: k-NN post-perturbation prediction (assessing drug embedding quality in isolation) and zero-shot perturbation retrieval (assessing alignment between cell and drug spaces). BiRD substantially outperforms chemical fingerprints (Morgan-FP) and small-molecule foundation model embeddings (ADMET-AI) on both tasks, with retrieval AUROC reaching 0.92 versus ~0.6 for baselines, and demonstrates strong cross-cell-line generalization.

## 2. Strengths

* The paper asks a well-scoped and important question: whether limitations in perturbation prediction stem from cell representations, drug representations, or their coupling and provides evidence that drug representation is a major bottleneck. This diagnostic framing is valuable independent of the specific method proposed and what I would highlight as the most noteworthy contribution.
* The experimental design is careful: reference cell lines used to construct BiRD are strictly excluded from all downstream evaluations, and the two evaluation tasks cleanly separate drug embedding quality (k-NN prediction) from alignment quality (zero-shot retrieval). Excellent baseline choices.
* The gene-level error decomposition with GO enrichment (Figure 3C) is a particularly nice analysis, providing functional interpretability for *why* BiRD outperforms alternatives: the enrichment for pathway-level processes (ribosome, cell cycle, adhesion) versus ADMET-AI's metabolism-associated terms is a concrete and biologically coherent finding.
* The scale of evaluation (45 cell lines, cross-cell-line transfer in Figure 5) is substantially larger than typical perturbation prediction benchmarks, and the consistent results across this breadth lend credibility to the claims.
* The paper is clearly written, well-structured, and appropriately scoped for a workshop contribution.

## 3. Weaknesses

**3.1 BiRD Requires Perturbation Data From Reference Cell Lines, Limiting Practical Applicability.**
The authors acknowledge this (Section 6) but somewhat understate its implications. BiRD embeddings are constructed from transcriptional responses in five reference cell lines, which are then appropriately excluded from any downstream fine-tuning sets. As I read it, this means that for any new drug, one must first screen it in reference contexts before generating its embedding. This is a fundamentally different operating regime than chemical-structure–based methods, which can embed any compound *in silico*. The strong performance gains likely reflect, in part, the fact that BiRD embeddings encode the very type of information (transcriptional response similarity) that the downstream tasks evaluate. The comparison is thus somewhat asymmetric: BiRD has access to biological response data while baselines do not. A fairer framing would explicitly characterize this as an upper bound or oracle condition, and discuss what fraction of the performance gap might be recoverable by structure-based methods with access to even limited response data (e.g., a hybrid approach).

**3.2 The k-NN Prediction Task May Overstate BiRD's Advantage.**
The k-NN task predicts a held-out drug's expression profile by averaging the profiles of its nearest neighbors in embedding space. Since BiRD is built from transcriptional response similarity, nearest neighbors in BiRD space are, by construction, drugs with similar transcriptional signatures. The spectral embedding is a dimensionality reduction of the Jaccard similarity matrix i.e., Euclidean distance in BiRD space reduces to an eigenvalue-weighted distance on the original drug similarity graph. The task therefore largely tests whether drugs with overlapping DE gene sets in reference cell lines have similar expression profiles in held-out cell lines, which is a question about cross-cell-line conservation of DE signatures more than embedding quality. It would be informative to compare against k-NN directly on the raw similarity matrix to isolate the contribution of the SVD step itself. This does not invalidate the result (the reference and evaluation cell lines are distinct), but the extent to which this structural advantage, rather than embedding quality per se, drives the observed gains would benefit from explicit discussion.

**3.3 The OT Alignment Objective is Evaluated Only as Retrieval, Not as Perturbation Prediction.**
The fine-tuning strategy (Section 3) aligns cell embeddings to drug embeddings, but the downstream evaluation is limited to retrieval (can we identify which drug perturbed a cell?) and clustering separability (Table 1). The paper does not evaluate whether the aligned embeddings improve actual perturbation *prediction* i.e., predicting the post-perturbation expression profile of a cell given its pre-perturbation state and a drug identity. This is the task most relevant to the drug discovery motivation stated in the introduction. Retrieval is a useful proxy, but the gap between "identify which drug was applied" and "predict what the drug will do to a new cell" is substantial. Similarly, the scIB separability analysis (Table 1) shows that fine-tuning improves drug-label separation, but the gap between BiRD (0.87 ± 0.02) and Morgan-FP (0.86 ± 0.02) is within reported standard deviations, suggesting the primary effect is fine-tuning itself rather than the choice of drug embedding. Whether the resulting embedding distances reflect meaningful biological relationships (e.g., mechanism of action) rather than arbitrary separability is not assessed.

**3.4 Sensitivity to BiRD Construction Choices is Not Explored.**
There are several design decisions in the BiRD pipeline: the number of reference cell lines (5), the signature size M, the significance threshold α, the embedding dimension L, and the choice of Jaccard over alternatives (e.g., cosine similarity on continuous log-fold changes). These could all influence downstream performance; no sensitivity analysis is provided. In particular, the choice to binarize DE signatures (top-M gene sets) discards magnitude information that could be informative; understanding the tradeoff between this simplification and robustness to noise would strengthen the contribution. Additionally, the authors describe the post-SVD drug embedding space as capturing "the intrinsic geometry of the drug embeddings" (Section 3), though this geometry is determined by the choice of similarity metric and spectral truncation rather than being an inherent property of the drugs themselves. The term is used more precisely in Section 4 ("intrinsic quality of the drug embedding space"); aligning the language would avoid implying that the SVD recovers a canonical drug manifold.

**3.5 Limited Foundation Model Diversity.**
All fine-tuning experiments use a single foundation model (BMFM). While the drug embedding comparison is the paper's primary contribution, the generality of the OT alignment strategy would be more convincing if demonstrated across multiple foundation models (e.g., scGPT, Geneformer), particularly given the documented variability in how different FMs respond to fine-tuning.

**3.6 Missing Meaningfulness Statement.**
The submission does not include a meaningfulness statement, which is required by the workshop guidelines. The camera-ready must include one.

## 4. Relation to Prior Work

The paper is well positioned relative to ChemCPA, CellOT, and BioLORD. The distinction from ChemCPA in fixing drug geometry a priori rather than learning it end-to-end, is clearly articulated and the decoupling is a methodological strength. The discussion would benefit from engaging more directly with the Connectivity Map lineage (Lamb et al., 2006; Subramanian et al., 2017), since the core idea of characterizing drugs by transcriptional response signatures and using signature similarity to relate compounds has a long history. BiRD can be seen as a principled geometric formalization of this intuition e.g. Jaccard on DE gene sets followed by spectral embedding, rather than cosine similarity on full expression signatures. Making this lineage explicit would better contextualize the contribution.

## 5. Questions for the Authors

1. How sensitive is BiRD performance to the number and identity of reference cell lines? Is there a minimum reference panel size below which performance degrades substantially?
2. Have the authors compared Jaccard similarity on binarized gene sets against continuous similarity measures (e.g., cosine similarity on log-fold change vectors)? What motivates the discretization?
3. Can the OT-aligned embeddings be used for actual perturbation prediction (predicting post-perturbation expression), rather than retrieval alone?
4. How does performance scale with the number of drugs in the candidate set for the retrieval task? At 350 drugs, near-chance baselines (~0.6 AUROC) suggest the task may not be highly discriminative.
5. Would a hybrid embedding that combines BiRD with chemical structure features further improve performance, or are the two signal sources largely redundant?
6. Have the authors considered supplementary metrics such as RMSE (to assess sensitivity to large per-gene errors) or Pearson/Spearman correlation on perturbation signatures (to assess whether the shape of the response profile is preserved, not just magnitude)?

## 6. Overall Assessment

This is a well-executed study that asks the right question and provides clear evidence that biologically grounded drug representations substantially improve perturbation modeling over structure-based alternatives. The experimental design is rigorous, the scale of evaluation is commendable (go Tahoe), and the gene-level functional analysis adds genuine interpretive value. The main limitations: the asymmetric comparison between BiRD (which requires screening data) and purely *in silico* baselines, the "induced" structure of the k-NN task, and the absence of actual perturbation prediction evaluation are real, but do not undermine the core finding. The paper makes a focused, well-supported contribution that is well suited for the LMRL workshop and is likely to inform how the community thinks about drug representation in perturbation modeling going forward. Recommending acceptance and looking forward to see it at the workshop.

---

### Official Review · Reviewer_FPwv · 2026-02-25
**Well-designed biological-response–derived drug embeddings for transcriptomics perturbation prediction and observation-to-drug retrieval**

**Rating:** 7
**Confidence:** 5

**Review:**

**Overview:**
The paper introduces biological-response–derived drug embeddings (BiRD) that characterize drug effects in transcriptomics assays. Similarity analysis of those embeddings allows for expression prediction subject to a drug by using expression subject to “similar” drugs.

They further implement a finetuning strategy that allows to align expression embeddings derived from transcriptomics foundation models with the corresponding drug embeddings via optimal transport matching. This allows retrieving the applied drug for unseen expression profiles in new cell lines.

The paper is well-written and the methodology is clearly explained. The method performs well compared to a set of baselines. Overall, the proposed method is novel and could be impactful in molecular screening for specific drug effects and predicting perturbation effects in new cellular contexts. As outlined later, some changes to the experimental evaluation and the addition of certain baselines could improve the impact further.


**Pros:**
- Clear description of the methodology
- The presented method for fusing drug effect representations into transcriptomics expression foundation model embeddings is elegant
- The paper includes compelling analysis of the observed functional enrichment compared to the usage of ADMET-AI embeddings that align with anticipated strengths and weaknesses.


**Areas of improvement:**
- The evaluation of Task 1 doesn’t establish the overall efficacy of the approach. Baselines for Task 1 quantify how well this specific approach (transfer between contexts) works using the particular choice of embeddings (BiRD) versus other molecular representations. To understand the overall impact, it would be interesting to also add a baseline that quantifies the “assay quality”, i.e., how close the method performs to the performance ceiling given the data. An example of such a baseline is the “experimental reproducibility” in [1].
- Identifying the performance driver: It is not entirely clear if the performance of the presented approach comes from having a relational fingerprint (BiRD) based on a drug-drug similarity matrix versus a structure-based fingerprint (Morgan) or if it is due to the specific design of the BiRD fingerprint. An interesting ablation would be to compare to other approaches of building the similarity matrix. For example, one could use cosine similarity between Morgan fingerprints or rely on existing knowledge graphs from the literature if available.
- A slight modification of the narrative and experimental design could improve the impact. On the surface, it is of limited use to retrieve perturbations that an experimental readout is subject to (Task 2) because we already know the drug that was applied (having run the experiment). However, the approach could be used to screen for drugs with similar effects in a few-shot setting, i.e., having access to a small percentage of different drugs per cellular context during training. This could be very impactful, allowing for sparser (and cheaper) experimental screening of molecules.


**Further comments:**
- It would be interesting to discuss how the training contexts were selected and how much that choice matters
- Inconsistent 350 drugs (line 18) vs. 380 drugs (line 69)
- There has been related work on contrastive learning with molecular representations and Phenomics data (images of perturbed cells) with similar use cases that would be worth discussing [2]
- An interesting future experiment would be to also evaluate the BiRD embeddings compared to the other fingerprints outside of the scope of transcriptomics prediction, for example, if they can be used to accurately predict ADMET properties using an MLP probe. This would however rely on the availability of such properties for the compounds observed in Tahoe.


[1] Wenkel, et al. "Txpert: Leveraging biochemical relationships for out-of-distribution transcriptomic perturbation prediction" Preprint 2025

[2] Fradkin, et al. "How molecules impact cells” NeurIPS 2024

---

### Official Review · Reviewer_hfFD · 2026-02-25

**Rating:** 7
**Confidence:** 5

**Review:**

The authors investigate a critical bottleneck in predicting cellular responses to small-molecule perturbations: the quality and nature of the drug representation. The paper hypothesizes that current shortcomings in single-cell foundation models stem from relying on chemically derived embeddings (like Morgan fingerprints or ADMET-AI) rather than biologically grounded ones.

To address this, the authors propose Biological-response-derived Drug Embeddings (BiRD), which are constructed by computing the Jaccard similarity of differentially expressed gene sets from reference cell lines and applying spectral decomposition. They also introduce an optimal transport (OT) objective to align the latent space of single-cell foundation models (specifically BMFM) with this new drug embedding space. Utilizing the large-scale Tahoe-100M dataset , the authors demonstrate that BiRD embeddings significantly outperform chemical and foundation-model baselines in both a k-NN post-perturbation prediction task and a zero-shot perturbation retrieval task.

### Strengths

- Scale of Evaluation: The use of the Tahoe-100M dataset (spanning ~50 cell lines and 350 drugs) represents a substantial leap over previous benchmarks like sci-Plex, providing a highly rigorous testing ground for transferability.

- Impressive Empirical Results: The performance gap between BiRD and the baselines is stark and compelling. Achieving an average AUROC of $0.92 \pm 0.04$ on zero-shot retrieval compared to $0.57 \pm 0.03$ for ADMET-AI is a massive improvement that validates the core hypothesis.

### Weaknesses

- Missing Baselines: While ADMET-AI and Morgan fingerprints are appropriate baselines for fixed representations, the paper discusses ChemCPA (which learns drug embeddings end-to-end implicitly) but does not empirically compare against it. An end-to-end baseline would help solidify whether explicit pre-computation of BiRD is strictly necessary.

---

### Meta-Review · Area_Chair_z5Vp · 2026-02-25

**Recommendation:** Accept (Poster)
**Confidence:** 4

**Metareview:**

Accept

---

### Decision · Program_Chairs · 2026-03-02

**Decision:**

Accept (Poster)

**Comment:**

Please see the meta-review.